# A Portable, Neurostimulation-Integrated, Force Measurement Platform for the Clinical Assessment of Plantarflexor Central Drive

**DOI:** 10.3390/bioengineering11020137

**Published:** 2024-01-30

**Authors:** Ashley N. Collimore, Jonathan T. Alvarez, David A. Sherman, Lucas F. Gerez, Noah Barrow, Dabin K. Choe, Stuart Binder-Macleod, Conor J. Walsh, Louis N. Awad

**Affiliations:** 1Department of Physical Therapy, Sargent College of Health and Rehabilitation Sciences, Boston University, Boston, MA 02215, USA; acollimo@bu.edu (A.N.C.);; 2Harvard John A. Paulson School of Engineering and Applied Sciences, Harvard University, Cambridge, MA 02138, USA; jonathanalvarez@g.harvard.edu (J.T.A.); lucasfgerez@gmail.com (L.F.G.); walsh@seas.harvard.edu (C.J.W.); 3Section of Rheumatology, Boston University, Boston, MA 02215, USA; 4Department of Physical Therapy, University of Delaware, Newark, DE 19716, USA; sbinder@udel.edu

**Keywords:** muscle strength, activation, burst superimposition, neuromuscular, stroke, gait

## Abstract

Plantarflexor central drive is a promising biomarker of neuromotor impairment; however, routine clinical assessment is hindered by the unavailability of force measurement systems with integrated neurostimulation capabilities. In this study, we evaluate the accuracy of a portable, neurostimulation-integrated, plantarflexor force measurement system we developed to facilitate the assessment of plantarflexor neuromotor function in clinical settings. Two experiments were conducted with the Central Drive System (CEDRS). To evaluate accuracy, experiment #1 included 16 neurotypical adults and used intra-class correlation (ICC_2,1_) to test agreement of plantarflexor strength capacity measured with CEDRS versus a stationary dynamometer. To evaluate validity, experiment #2 added 26 individuals with post-stroke hemiparesis and used one-way ANOVAs to test for between-limb differences in CEDRS’ measurements of plantarflexor neuromotor function, comparing neurotypical, non-paretic, and paretic limb measurements. The association between paretic plantarflexor neuromotor function and walking function outcomes derived from the six-minute walk test (6MWT) were also evaluated. CEDRS’ measurements of plantarflexor neuromotor function showed high agreement with measurements made by the stationary dynamometer (ICC = 0.83, *p* < 0.001). CEDRS’ measurements also showed the expected between-limb differences (*p*’s < 0.001) in maximum voluntary strength (Neurotypical: 76.21 ± 13.84 ft-lbs., Non-paretic: 56.93 ± 17.75 ft-lbs., and Paretic: 31.51 ± 14.08 ft-lbs.), strength capacity (Neurotypical: 76.47 ± 13.59 ft-lbs., Non-paretic: 64.08 ± 14.50 ft-lbs., and Paretic: 44.55 ± 14.23 ft-lbs.), and central drive (Neurotypical: 88.73 ± 1.71%, Non-paretic: 73.66% ± 17.74%, and Paretic: 52.04% ± 20.22%). CEDRS-measured plantarflexor central drive was moderately correlated with 6MWT total distance (r = 0.69, *p* < 0.001) and distance-induced changes in speed (r = 0.61, *p* = 0.002). CEDRS is a clinician-operated, portable, neurostimulation-integrated force measurement platform that produces accurate measurements of plantarflexor neuromotor function that are associated with post-stroke walking ability.

## 1. Introduction

Full volitional access to the force-generating capacity of muscle is a hallmark of unimpaired neuromotor function and can be measured clinically as the central drive ratio (i.e., the ratio of the forces produced without and with superimposed electrical stimulation [1,2]). The maximum force produced without stimulation is the muscle’s voluntary strength capacity (MVC), whereas the maximum force produced with electrical stimulation is the muscle’s maximum force-generating ability (MFGA). In neurotypical adults, the MVC force approximates the MFGA force, resulting in a central drive ratio measurement of ~1.0, which is interpreted as full volitional activation of the muscle’s innate force-generating capacity. In contrast, central drive ratios below 1.0 indicate a deficit in central drive to muscle [3,4,5].

Muscle strength is classically assessed using tests of voluntary force production; however, muscle weakness may stem from reduced force-generating capacity (i.e., reduced MFGA), an inability to access latent muscle capacity (i.e., reduced central drive), or a combination of these deficits. Identification of the nature of a patient’s muscle weakness is necessary to match patients with targeted interventions. Indeed, in musculoskeletal rehabilitation, reduced central drive to muscle is a well-documented cause of weakness after injury or surgery [6,7,8], and specific therapeutic interventions are required to alleviate persisting strength deficits [9]. Though the underlying cause for reduced central drive may be different in neurological diagnostic groups, emerging evidence suggests a similar need for identification of the nature of muscle weakness to guide the prescription of targeted interventions [4,10,11]—such as for plantarflexor muscle weakness after stroke.

In people post-stroke, the paretic plantarflexor muscles are documented to be 55% weaker than the plantarflexor muscles of neurotypical controls [12], with this weakness stemming from reductions in both force-generating capacity (i.e., MFGA) and central drive [3,4,5,13]. Further, the magnitude of plantarflexor central drive impairment is strongly associated with asymmetry in the propulsive forces generated by the paretic and non-paretic limbs during walking, a key gait impairment after stroke [4]. Diagnostic approaches that can elucidate the extent to which a specific patient’s plantarflexor weakness is the result of reduced capacity (i.e., MFGA), requiring strength training, or reduced central drive, requiring neuromodulatory intervention, are critically needed to advance clinical decision making informed by an awareness of the exact nature of the strength impairment.

Whereas gross strength is routinely measured in clinical settings, measurement of central drive is primarily carried out in research settings. Barriers complicating the acquisition of central drive measurements have hindered clinical uptake and applications. Namely, excessive cost, lack of physical space, and difficulty transferring patients with mobility impairments onto and off the measurement devices are factors that make gold-standard stationary dynamometers not suitable for routine clinical measurement of central drive [14,15]. Affordable and portable alternatives, such as hand-held dynamometers, are accompanied by methodological concerns due to variability in setup [16]. Furthermore, neither stationary nor hand-held dynamometers typically have the integrated real-time force data and electrical stimulation needed to assess central drive reliably and accurately. Therefore, a critical first step in increasing clinical access to plantarflexor central drive measurements is the development of neurostimulation-integrated plantarflexor force measurement systems that are physically accessible by individuals with mobility impairments and compatible with the physical space restrictions of most clinical settings.

The development of a measurement system that enables the routine clinical assessment of plantarflexor central drive would constitute a powerful diagnostic and clinical decision-making tool for stroke rehabilitation. However, limitations in the measurement itself must also be considered. More specifically, because electrically evoked muscle contractions may not fully activate the residual force-generating capacity of the muscle [1,17], central drive measurements that depend on electrical stimulation of muscle often underestimate a muscle’s true MFGA and overestimate its central drive. To overcome this limitation, gold-standard, laboratory-based measurement approaches have developed and use adjustment equations to calculate true central drive from measured central drive [13,17]. Because adjustment equations are inherently specific to the equipment and methods used, device-specific adjustment equations from collected user data are thus required for the development of new, clinic-based, plantarflexor central drive measurement devices.

The objective of this study was to develop and validate a clinically accessible plantarflexor force and central drive measurement system (CEntral DRive System [CEDRS]). Our validation of the CEDRS device consisted of two experiments. The first evaluated the accuracy of its plantarflexor force measurements in neurotypical adults, and the second evaluated its ability to measure plantarflexor central drive deficits in people with post-stroke hemiparesis. We hypothesized that plantarflexor MVC force measurements made by CEDRS would be accurate compared to MVC force measurements made by a gold-standard stationary isokinetic dynamometer (HUMAC Norm, Computer Sports Medicine Inc., Stoughton, MA, USA). Moreover, we hypothesized that we would be able to derive a well-fit device-specific adjustment equation for CEDRS from the relationship between true central drive and device-measured central drive. Finally, we hypothesized that CEDRS would produce valid measurements of plantarflexor strength and central drive as evidenced in their (1) reflection of the known asymmetry across paretic and non-paretic limbs, (2) demonstration of impairment in both limbs compared to the limbs of neurotypical individuals, and (3) association with established metrics of walking function derived from the six-minute walk test, which is the gold-standard functional measure of post-stroke walking ability.

## 2. Materials and Methods

### 2.1. Device Development

#### 2.1.1. Hardware

CEDRS is a stand-alone device that weighs approximately 18 lbs. and can be securely strapped to any standard medical examination bed (Figure 1A). CEDRS is designed to measure isometric plantarflexion torque via a hardware platform that is portable and allows for adjustability across individuals, while retaining structural rigidity to enable accurate torque measurements. Together, these design elements allow CEDRS to be a clinically accessible, yet accurate and valid system for measuring plantarflexor neuromotor function.

CEDRS consists of two main components: a distal assembly containing a footplate and torque load cell and a proximal fixture of two parallel aluminum extrusions with adjustable shank and thigh supports to align and secure the participant’s leg during testing. Both components were designed to maximize rigidity to ensure consistent kinematics across participants and minimize variability during testing within a given participant.

The design of the distal assembly was inspired by existing isokinetic dynamometer designs, such as the HUMAC Norm or Biodex, which are the gold standard for measuring joint torques [15]. It consists of a stainless-steel footplate supported at one end by a pin joint and connected on the other end to a static reaction torque sensor (ATO-TQS-S04, ATO, Diamond Bar, CA, USA) through a stainless-steel shaft, supported by an additional pin joint, to measure plantarflexion torque. The torque load cell has a rated maximum of 110.6 ft-lbs. (with a 166 ft-lb. safety overload). The footplate is fixed at a neutral ankle angle (90°). A heel cup attached to the footplate helps support the weight of the participant’s foot. Additionally, slots in the footplate provide anchoring points for a series of straps used to secure the foot to the footplate during testing.

The proximal fixture is composed of two parallel aluminum extrusions (25 Series, 80/20 Inc., Columbia City, IN, USA) extending proximally from the distal assembly with additional 3-D printed supports for the shank and thigh and eye bolts to secure an over-the-shoulder harness. These features serve to align, support, and constrain the participant during testing. To prevent translation of the device relative to the medical examination bed, the entire device is secured to the bed with ratchet tie-down straps anchored to slots along the device. The device is calibrated with a torque wrench (Husky 56394, Home Depot Inc., Cobb County, GA, USA) to obtain the torque to voltage calibration.

#### 2.1.2. Software

A custom electronics box houses the signal conditioning hardware, load cell amplifier (ATO-LCTR-OA, ATO, Diamond Bar, CA, USA), and a Raspberry Pi (Raspberry Pi 4 Model B, Raspberry Pi Foundation, Cambridge, UK), which is used to sample and provide a real-time display of generated torque. Running continuously on the Raspberry Pi is a PyQt-based graphical user interface (GUI) [18], which leverages PyQtGraph [19] for real-time torque visualization. Additionally, specific torque thresholds can be overlaid on the GUI in real time for biofeedback. The generated torque is sampled at 1 kHz by using an external data acquisition unit (DAQ; Powerlab 8/35, ADInstruments, Dunedin, New Zealand) for post-processing.

#### 2.1.3. Integrated Stimulator

A commercial electrical stimulation device (RehaMove 3, Hasomed GmbH, Magdeburg, Sachsen-Anhalt, Germany) is used to deliver electrical stimulation to participants during central drive testing. A custom GUI was written to interface with the stimulation device, allowing researchers to tailor stimulation to each participant by modulating amplitude, frequency, duration, and number of pulses of the electrical stimulus and to manually trigger the device. The GUI also enables real-time syncing of force and electrical stimulation data, to enable proper measurement of force prior to and following the burst.

### 2.2. Experiment #1: Device Accuracy and Adjustment Equation Development

Study participants completed one of two experiments. The first experiment included neurotypical individuals and aimed to assess the accuracy of CEDRS’ measurements of plantarflexor force production and develop a CEDRS-specific adjustment equation for central drive measurements.

#### 2.2.1. Study Procedures

Twenty-three adults (>18 years old) who reported no diagnosis of a neurological condition and had no observable gait deficits were recruited for this study. The sample size for this experiment was determined via an *a priori* power analysis [20]; expecting intraclass correlation coefficients (ICC) ≥ 0.8 [21,22,23] with confidence tolerance of 0.2, α = 0.05, and β = 0.80 and two repeated measures (k = 2), a minimum of 12 participants were needed (R Version 4.0.4, ICC.Sample.Size). Each study participant signed informed consent forms approved by the Boston University Institutional Review Board.

##### Participant Setup

Study participants performed procedures using both the stationary dynamometer and CEDRS in a randomized order. Participants lay supine with their knee fully extended and ankle in a neutral position. Straps were used to stabilize the foot and leg and an over-the-shoulder harness was used to prevent movement during contractions. A monitor displayed real-time visual feedback of the torque to the participant. Two self-adhesive surface electrodes (CFF305, Axelgaard Manufacturing Co., Ltd., Fallbrook, CA, USA) were placed over the plantarflexor muscles and used to deliver electrical stimulation during the tests; one placed proximally over the widest part of the gastrocnemius, and one placed over the distal soleus muscle belly [4].

##### Selecting Burst Parameters

The burst superimposition test was used for all central drive assessments. During central drive testing, a fixed-amplitude supramaximal burst (150 ms, 100 Hz, 150 mA; Figure 1A) was manually applied by the integrated stimulator when plantarflexor torque reached a visual steady state. The pulse duration of the burst was individualized for each participant based on maximization of their twitch torque response to sequential stimulation at rest (see Figure 1B). The burst superimposition testing protocol that was used to measure central drive for each participant was preceded by (1) a single maximum voluntary contraction (without stimulation) to potentiate the muscle and (2) a pulse duration ramp protocol to identify the optimal pulse duration for each participant. In brief, the pulse duration ramp protocol consisted of the integrated stimulator delivering repeated twitch stimulations at rest with progressively increasing pulse widths (in 50 µs increments) that ranged from 50 µs to 600 µs. The minimum pulse width that generated the largest twitch response (i.e., largest torque output) was selected for each participant and used for the remainder of testing; the range of pulse widths ultimately selected across participants in this study was between 350 and 600 µs. Pulse duration optimization was completed on the device that was randomized to be tested first.

##### Central Drive Testing

Study participants completed seven central drive tests on both the stationary dynamometer and CEDRS, with each test separated by a minimum of two minutes to minimize fatigue. During the first central drive test, study participants were instructed to maximally contract their plantarflexor muscles (i.e., produce their MVC torque) prior to delivery of the stimulation burst used to determine the MFGA of the plantarflexors. MFGA was defined as the maximum torque elicited following application of the stimulation burst. (Note: Although both the stationary dynamometer and CEDRS measure joint torque, throughout this manuscript, we use joint torque as a proxy for plantarflexor force, as the lever arm remains fixed and the joint center is aligned with the load cell). If a participant was unable to volitionally produce 95% of their MFGA torque, a target line was displayed on the monitor for feedback and the test was repeated up to three times. If the study participant was unable to voluntarily achieve 95% of their MFGA torque in three trials, the participant was dismissed due to inability to achieve a true estimation of the MFGA.

The MVC test(s) were followed by two submaximal voluntary contractions in a randomized order at each: 25%, 50%, and 75% of the MFGA measured during the MVC central drive test (%MFGA). After the completion of all tests on the first device, a ten-minute rest was provided before beginning testing on the second device. If, on the second device, volitional torque of 95% was not reached during the MVC trial, additional rest was provided to minimize fatigue. All participants who were able to reach 95% volitional torque on the first device, were also able to do so on the second device within three attempts.

#### 2.2.2. Data Processing

Trials where voluntary torque output (F_vol_) had a variance greater than 0.1 were flagged and visually assessed by at least two investigators to confirm F_vol_ was at a steady state when stimulation was delivered. Trials where stimulation was delivered when F_vol_ was increasing or decreasing were excluded, as motor unit recruitment and decruitment can influence post-burst potentiation, and thus central drive [24].

Torque data were low-pass filtered at 10 Hz using a fourth-order Butterworth filter using custom MATLAB scripts. For each trial, F_vol_ was calculated as the average torque over the 100 ms prior to the burst stimulation, and stimulation-elicited force (F_stim_) was calculated as the peak torque following burst stimulation.

MVC was defined as the F_vol_ produced during the single 100% trial conducted on each device. To determine the accuracy of CEDRS’ measurements, plantarflexor MVC measurements across the HUMAC Norm and the CEDRS device were compared. To develop an adjustment equation, *device-measured* and *true* central drive were computed for all seven trials on the CEDRS device. *Device-measured* central drive was computed as the ratio of F_vol_ to F_stim_. *True* central drive was computed as the ratio of F_vol_ to participant-specific MFGA, where MFGA was equal to the maximum torque reached during the MVC test. The adjustment equation was constructed to map the *device-measured* central drive to the *true* central drive.

#### 2.2.3. Statistical Analyses

To determine the accuracy of plantarflexion MVC torque acquired using CEDRS versus the gold-standard stationary dynamometer, intra-class correlation (ICC_2,1_) coefficients [25] were calculated. Additionally, limits of agreement were calculated using the Bland–Altman method.

To develop the adjustment equation, a third-order polynomial relationship was used to relate *device-measured* and *true central drive* (F_vol_/MFGA) for each of the sub-maximal trials completed on each device [13]. To address the theoretical understanding that a device-measured central drive of 0 should result in a true central drive of 0, the equation was fixed at the origin. The resultant adjustment equation was then tested using leave-one-out cross-validation to verify the overall fit and error of the equation were stable across all participants. In other words, this validation confirmed that no single participant was significantly impacting the equation fit. Then, to ensure the residual error of the adjustment equation was consistent across all force levels, we assessed the heteroscedasticity of the residuals using the Bruesch–Pagan test.

In all analyses, alpha was set to 0.05 and means ± standard deviations are reported for all variables. Statistical analyses were calculated using SPSS (version27, IBM Corp, Armonk, NY, USA) and MATLAB (R2022a).

### 2.3. Experiment #2: Post-Stroke Evaluation

The second experiment added individuals with post-stroke hemiparesis and aimed to validate CEDRS’ measurements of plantarflexor neuromotor function by (1) evaluating differences between the paretic and non-paretic limbs, (2) comparing these post-stroke measurements to measurements made in neurotypical participants, and (3) examining their association with established metrics of post-stroke walking function.

#### 2.3.1. Study Procedures

Twenty-six individuals post-stroke were recruited for this study and signed informed consent forms approved by the Boston University Institutional Review Board. The sample size was all participants available at the time of analysis, with N = 26 considered sufficient for the experimental aims. Inclusion criteria were >18 years old, at least 6 months post-stroke, able to achieve a neutral ankle angle, and able to communicate clearly with investigators. Additionally, participants had to be able to walk independently with or without use of an assistive device that was not designed to restrict ankle plantarflexion (e.g., solid ankle-foot orthoses were not allowed during all testing conditions). Exclusion criteria included sub-cortical stroke, score of >1 on question 1b and >0 on question 1c on the NIH Stroke Scale (to screen for cognitive impairment), pain that impairs walking, neglect and hemianopia, unexplained dizziness in the last 6 months, or more than two falls in the last month.

##### Central Drive Testing

Participant set-up and pulse width optimization procedures from Experiment #1 were repeated for the post-stroke group using the CEDRS. Participants then completed 3 MVC burst superimposition trials for each leg (see Experiment #1 description). In brief, the supramaximal burst (150 ms, 100 Hz, 150 mA) was applied when plantarflexor torque reached a visual steady state. A minimum of two minutes of rest were provided between tests. Testing was performed on the paretic leg followed by the non-paretic leg. If a quantifiable decrease in MVC force prior to burst delivery was seen by the investigators, the test was excluded and an additional test was completed following a longer rest period.

##### Six-Minute Walk Test Procedures

All post-stroke participants also completed a six-minute walk test (6MWT) that was administered and scored by a licensed physical therapist. During the 6MWT, a research assistant recorded the distance covered during each minute. The total distance covered during the 6MWT is considered a key predictor of post-stroke community walking activity [26]. More recently, assessing whether or not an individual speeds up or slows down during the test (i.e., the distance-induced change in six-minute walk test speed; %Δ6MWT speed) has been shown to improve prediction of community walking activity, beyond using the 6MWT distance alone [27]. The %Δ6MWT speed was calculated, as a percentage, as the difference in distance walked in the first versus sixth minutes, divided by the distance walked in the first minute. Both 6MWT distance and %Δ6MWT speed were used to quantify post-stroke walking function in this study.

#### 2.3.2. Data Processing

As described in Experiment #1, trials were visually assessed and flagged for high variance or if the stimulation was delivered while force was changing. Additional post-stroke trials were flagged and visually inspected if F_vol_ was less than 90% of the within-trial peak prior to the burst (i.e., participants reached a peak and quickly dropped off); this procedure helped ensure that the stimulation was superimposed on a maximum voluntary contraction, rather than a sub-maximal steady force.

F_vol_ and F_stim_ were averaged across the successful central drive tests, and these averages were used to calculate *device-measured* central drive, for both the paretic and non-paretic limbs. *Adjusted* central drive was computed for each limb using the adjustment equation derived from experiment #1. To compare the central drive of the paretic and non-paretic limbs to neurotypical individuals, *adjusted* central drive was also calculated for the MVC trials in experiment #1.

#### 2.3.3. Statistical Analyses

Our validation of CEDRS first tested the prediction that CEDRS’ measurements of plantarflexor neuromotor function would be different across paretic, non-paretic, and neurotypical limbs. More specifically, one-way ANOVAs with Sidak post hoc corrections were used to compare maximum voluntary plantarflexor strength (Average F_vol_), plantarflexion strength capacity (Average F_stim_), and adjusted central drive across limbs. Eta-squared effect sizes with 95% confidence intervals were used to describe the magnitude of differences and were interpreted as small (≥0.01), medium (≥0.06), or large (≥0.14) effects. Our validation of CEDRS secondarily tested the prediction that CEDRS’ measurement of paretic central drive (a promising biomarker of post-stroke neuromotor impairment [4]) would be associated with post-stroke walking function. More specifically, bivariate correlations were used to evaluate the association between paretic central drive and the 6MWT distance and the distance-induced %Δ6MWT speed. Consistent with Experiment #1, alpha was set to 0.05, means ± standard deviations are reported for all variables, and statistical analyses were calculated using SPSS.

## 3. Results

### 3.1. Experiment #1: Device Accuracy and Adjustment Equation Development

#### 3.1.1. Participants

Twenty-three neurotypical adults (11 males, 27 ± 3 years old, 19 right dominant) enrolled in the study. Seven participants were excluded: four for not achieving the 95% MFGA target, two for generating >110.6 ft-lbs. of plantarflexion force (the device limit), and one for opting to stop testing. Data from the remaining 16 individuals (7 males, 27 ± 3 years old, 13 right dominant) were available to evaluate maximum voluntary strength across the two devices. After excluding trials where F_vol_ was not at a steady state (see 2.2.2 Data Processing), an average of 13.63 ± 1.05 out of 14 possible tests were available for each participant (14 usable tests for 14 participants, 12 usable tests for 1 participant, and 10 usable tests for 1 participant).

#### 3.1.2. Accuracy of Force Measurement

The average maximum voluntary contraction torque was found to be 73.93 ± 11.46 ft-lbs. on the HUMAC Norm and 77.47 ± 12.56 ft-lbs. on the CEDRS device. There was a high degree of agreement between the two devices (ICC = 0.83 [0.549, 0.939], R^2^ = 0.739, *p* < 0.001, Figure 2A). The magnitude of mean difference between devices was 3.70 ± 6.64 ft-lbs (Figure 2B).

#### 3.1.3. Development and Validation of Adjustment Equation

Our development of an adjustment equation specific to the CEDRS device resulted in a third-order polynomial mapping *device*-measured central drive (x) to adjusted central drive.
Adjusted Central Drive = 0.798x^3^ − 0.772x^2^ + 0.868x(1)

This equation was well fit (R^2^ = 0.913, RMSE = 7.2%) for explaining the variance between measured central drive and true central drive (Figure 3). The 95% confidence intervals for each coefficient were [0.794, 0.803], [−0.778 −0.766], and [0.866, 0.870], respectively. The Breusch–Pagan Test characterized the homoscedasticity of the residuals of the adjustment equation (*p* = 0.996), meaning error does not systematically change based on the magnitude of device-measured central drive. Leave-one-out cross-validation resulted in an RMSE of 7.2 ± 0.16%, confirming equal error across all the participants.

### 3.2. Experiment #2: Post-Stroke Evaluation

#### 3.2.1. Participants

Twenty-six individuals (19 males, 61 ± 10 years old, chronicity 6 ± 4 years, 11 right paretic) completed the additional testing required for experiment #2. Data for the 6MWT were unavailable for two participants who were unable to complete the test due to elevated heartrates. Additionally, paretic limb central drive data were unavailable for one participant and non-paretic central drive data were unavailable for a different participant; although tests were completed with these participants, F_vol_ was deemed to not be at a steady state in post-processing.

#### 3.2.2. Feasibility of Using CEDRS Post-Stroke

All twenty-six individuals post-stroke were able to use CEDRS to collect plantarflexor force measurements. A physical therapist assisted the participants into and out of the device, primarily by helping the participant maintain their balance when sitting onto and standing up from the clinical examination table and aligning the participant’s foot with CEDRS’ heel cup. An aide helped with securing the study participant to the device with CEDRS’ integrated straps, but did not offer critical support (i.e., operation of CEDRS requires only one person).

#### 3.2.3. Evaluation of Deficits Post-Stroke

After excluding trials where F_vol_ was not at a steady state (see 2.3.2 Data Processing), an average of 2.42 ± 0.84 out of three tests was available per participant for the paretic limb (three usable tests for 16 participants, two usable tests for 6 participants, and one usable test for 3 participants, zero usable tests for 1 participant). Similarly, 2.54 ± 0.75 out of three tests for the non-paretic limb were available (three usable tests for 17 participants, two usable tests for 7 participants, one usable test for 1 participant, and zero usable tests for 1 participant). Test usability varied across the paretic and non-paretic limbs. F_vol_, MFGA, and central drive for neurotypical, paretic, and non-paretic limbs are listed in Table 1. There were significant differences, with large effect sizes, among groups for F_vol_ (*p* < 0.001, η^2^ = 0.574 [0.397, 0.674]), MFGA (*p* < 0.001, η^2^ = 0.474 [0.280, 0.593]), and central drive (*p* < 0.001, η^2^ = 0.403 [0.205, 0.534]). Post hoc comparisons revealed lower paretic values for all variables compared to the neurotypical limbs (*p* < 0.001 for all) and non-paretic limbs (F_vol_
*p* < 0.001; MFGA *p* = 0.001; central drive *p* = 0.001). Compared to the neurotypical limbs, non-paretic limbs also had lower F_vol_ (*p* < 0.001), MFGA (*p* < 0.001), and central drive (*p* = 0.008; Figure 4A,B).

Validation data for the 6MWT were available for 23 of the 26 study participants; 2 participants did not have 6MWT data, and 1 participant did not have any successful paretic central drive trials. On average, participants walked a total distance of 321.09 ± 103.34 m, with 9 participants increasing speed between the 1st and 6th minute, 13 slowing down, and 2 maintaining their speed. Paretic limb central drive was significantly associated with 6MWT distance (r = 0.685, *p* < 0.001; Figure 4C) and distance-induced %Δ6MWT speed (r = 0.611, *p* = 0.002; Figure 4D).

## 4. Discussion

The CEDRS measurement system is a portable plantarflexor force measurement device with an integrated electrical stimulator and graphical user interface that enables the assessment of plantarflexor central drive in clinical settings. The findings of this study demonstrate that CEDRS is usable by clinicians and patients post-stroke; can measure plantarflexor strength with similar accuracy to gold-standard stationary dynamometers, while requiring a fraction of the space and cost; and produces valid measurements of plantarflexor neuromotor function that are associated with known post-stroke deficits.

Of primary importance is CEDRS’ accurate measurement of force relative to gold-standard dynamometry. The high degree of agreement between MVC torque across devices can likely be attributed to accurate torque calibration and similarities between user positioning within the device (i.e., supine with ankle immobilized at 0° dorsiflexion). With high reliability between devices and limits of agreement that crossed zero, differences in the force measurements made by using the gold-standard dynamometer and CEDRS were far below the previously reported minimum detectable change of 22.6 ft-lbs. for plantarflexor force [22]. Small measurement errors between MVC contractions will always exist, as MVCs are inherently variable [28]. This is to say that, even in the presence of a perfect device, small fluctuations in MVC are expected across measurement trials due to muscle physiology changes or fluctuations in volitional effort. To address this potential limitation, we ensured adequate rest and provided visual biofeedback with goal setting across all trials. Although small, measurement error may also stem from slight differences in ankle inversion across systems; the foot plate on the HUMAC Norm is slightly inverted (~16°, HUMAC Norm, Computer Sports Medicine Inc., USA) relative to the new device.

With respect to the measurement of central drive, theoretical and methodological limitations requiring an adjustment equation to correct for the overestimation of central drive that occurs due to incomplete electrically induced contractions are well known [1,17]. To address these limitations, systematic error can be mapped and modeled using neurotypical samples where measurement of MFGA is more reliable and then applied to clinical populations [13]. In experiment #1, a third-order polynomial adjustment equation explained a high degree of the variance between measured central drive and true central drive. Previously, a second-order polynomial has been used to adjust voluntary activation in the quadriceps [7,17] and a third-order polynomial has been used to adjust voluntary activation in the plantarflexors [13]. Despite subtle differences in approach, our equation explains 91% of the variance while ensuring no theoretical violations in the correction (i.e., fixing the y-intercept at zero is necessary to prevent inaccurate adjustment at the lowest levels of volitional activation). Additionally, we found a relatively small degree of additional error in leave-one-out cross-validation, confirming the CEDRS’ adjustment equation was not heavily influenced by participant-specific factors, including variations in pad placement or positioning.

An important contribution of this work is the demonstrated feasibility in using CEDRS with individuals post-stroke. The vast majority of participants successfully completed central drive testing with CEDRS on both limbs and were able to transfer onto and off the device with minimal assistance. It is especially noteworthy that adjusted central drive values from these tests were similar to prior results using the burst superimposition test with a stationary dynamometer (KIN-COM III, Chattecx Corp, Chattanooga, TN, USA). Indeed, CEDRS measurements taken across the paretic and non-paretic limbs of the 26 post-stroke participants are similar to a previous study of 40 people post-stroke with similar inclusion and exclusion criteria, with respect to adjusted central drive in the paretic (50% in the current study vs. 51%) and non-paretic limbs (70% vs. 70%), as well as measurement of large interlimb differences in plantarflexor MVC (−42% vs. −40%), MFGA (−27% vs. −19%), and central drive (−28% vs. −27%) [4]. These similarities suggest that clinicians can use CEDRS and the accompanying adjustment equations to accurately measure plantarflexor central drive.

Ultimately, investigating the clinical and rehabilitative implications of plantarflexor central drive deficits in individuals post-stroke is an emerging area of research. Indeed, to our knowledge, this is the first study to directly compare plantarflexor MVC, MFGA, and central drive from individuals post-stroke to a neurotypical control group and to evaluate the relationship between paretic limb central drive and the gold-standard clinical assessment of walking function, the 6MWT. Our findings suggest that central drive may be a promising biomarker of post-stroke neuromotor impairment. Indeed, CEDRS-measured plantarflexor central drive reflected the known asymmetry between the paretic and non-paretic limbs, distinguished non-paretic limbs from neurotypical limbs, and was associated with walking function. Critically, unlike non-specific measures of post-stroke impairment (like the total distance walked during the 6MWT), the clinical assessment of plantarflexor central drive elucidates the primary nature of an individual’s plantarflexor weakness; the specificity of this metric creates new opportunities for targeted clinical intervention. Further exploration of the relationship between individuals’ central drive and response to mechanistically different gait restorative treatments (e.g., soft robotic exosuits and functional electrical stimulation) is necessary to establish the prognostic value of plantarflexor central drive measurements post-stroke.

### Limitations

There are a few limitations to CEDRS, most notably the maximum torque measurement of 110.6 ft-lbs. The decision to use this load cell was a balance of size and function. The next size load cell would enable testing up to 368.78 ft-lbs., but would also approximately double both the size and weight (ATO-TQS-S01, ATO, Diamond Bar, CA, USA) of CEDRS. However, post-stroke individuals’ average plantarflexion strength capacity of the stronger, non-paretic limb is approximately 113.6 ft-lbs. [5]. Thus, the chosen load cell will work for most post-stroke participants. Furthermore, this was not an issue in our current study, as we saw an average plantarflexion strength capacity of 64 ft-lbs. on the non-paretic limb, which is significantly lower than that measured in previous work. Additionally, only two neurotypical individuals were excluded for exceeding the maximum torque rating. A larger load cell may be needed for working with stronger participants.

A second limitation is that the footplate is fixed at a neutral angle. This requires that participants can achieve a neutral angle in dorsiflexion, which can often be difficult after a stroke. Future iterations of CEDRS should include an adjustable angle footplate, to enable testing with individuals with more restricted ankle movement.

A limitation of the study design is that neurotypical and post-stroke cohorts were not age-matched. While age-matching groups was not required to develop and validate the CEDRS device and does not impact the primary results of this study, our comparison of post-stroke and neurotypical central drive does not account for known reductions in central drive due to aging [29]. Additionally, while this work demonstrates the accuracy of CEDRS for neurotypical plantarflexor strength and central drive measurements, as well as the validity of measurements from post-stroke individuals, it does not compare measurements between CEDRS and the gold-standard stationary dynamometer for individuals post-stroke. This decision was made to reduce the number of central drive tests participants post-stroke had to complete, but remains as a limitation of this study. In particular, the range of plantarflexion strength for individuals post-stroke is markedly lower than neurotypical adults; the accuracy of MVC measurements using CEDRS at lower forces should be evaluated with additional testing. Moreover, we acknowledge limited female participants in our clinical cohort, due to limitations in recruitment and our participant pool. In future studies, we aim to involve a more heterogeneous clinical cohort to better assess the applicability of our findings across a broader demographic.

Future work should also continue to improve clinical usability of central drive measurements. For example, although common in laboratory assessment, manual triggering of the superimposed burst presents a usability concern for clinical adoption. In fact, some trials herein were removed from analysis due to stimulation being delivered during periods of linear force increases or decreases. The next step for CEDRS is to refine a paradigm to automate delivery of burst stimulation, based on the force profiles observed in this study. Additionally, while only one physical therapist was needed to aid the participants when getting into the device, a second person was needed to help manage foot strapping. The next iteration of the system should improve the design to reduce overall personnel needs. Finally, in this study, the device was successfully used by trained research physical therapists, but future work should test the usability of this system in clinical settings and improve the GUI and system as needed, based on clinician feedback.

While this study implements the adjustment equation in a post-stroke population, more targeted validation testing should be performed to confirm usability of this equation in clinical populations. Previous equations have been successfully validated in ACL injury and post-stroke populations, and we would expect to see the same here, especially given the similarity in our central drive results for paretic and non-paretic limbs to prior studies. Moreover, due to the nature of how the adjustment equation was created, adjusted central drive values cannot exceed 89%, even with a measured central drive of 100%. This means that for high-performing participants, we cannot know where in the range of 89–100% someone’s true central drive is. An alternative equation or adjustment process may be needed for higher-level stroke participants with almost fully intact central drive.

## 5. Conclusions

The diagnostic and prognostic potential of routine clinical measurement of plantarflexor central drive can only be realized if the technological limitations that have hindered widespread clinical adoption are addressed. Namely, excessive cost and space requirements of stationary dynamometers and inaccuracies of hand-held dynamometers, prevent routine clinical measurement of central drive. We developed CEDRS, a portable neurostimulation-integrated plantarflexor force measurement system, to make central drive measurements accessible in clinical settings. The findings of this study demonstrate that CEDRS can accurately measure plantarflexor strength and central drive deficits in neurotypical adults and after stroke. CEDRS has the potential to advance point-of-care neuromotor diagnostics that facilitate advanced clinical decision making informed by quantitative measurements of neuromotor function.

## Figures and Tables

**Figure 1 bioengineering-11-00137-f001:**
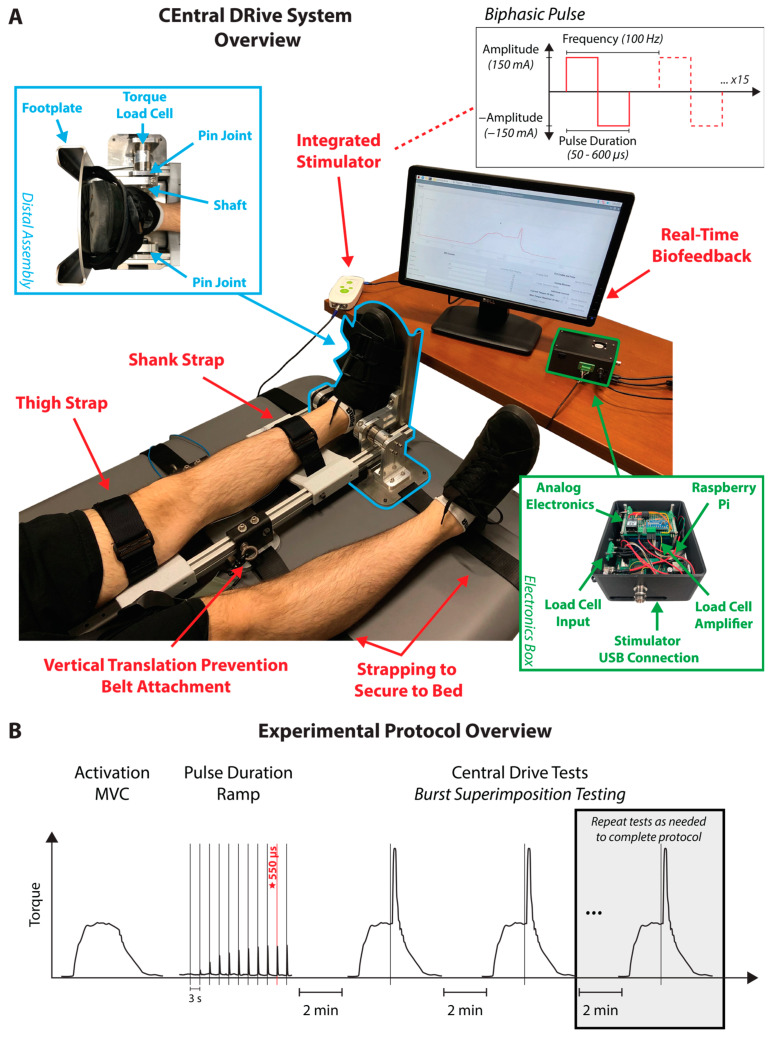
**Overview of CEDRS** (**A**) CEDRS is a portable central drive measurement system that can be attached to any standard evaluation bed. The foot, shank, and thigh are secured to the device with strapping, and an over-the-head harness prevents vertical translation of the participant during testing. An integrated electronics box stores data, provides real-time feedback, and integrates electrical stimulation signals with torque readings. The integrated stimulation applies 15 biphasic pulses at 150 mA and 100 Hz during central drive tests; pulse width was individualized for each participant. (**B**) The testing protocol begins with a maximum voluntary contraction (MVC) to potentiate the muscle, which is immediately followed by a pulse duration ramp that is used to identify the optimal pulse duration for each participant. In brief, during the pulse duration ramp, the pulse duration of the twitch stimulation is progressively increased from 50 µs to 600 µs by 50 µs increments. The pulse duration resulting in the largest twitch response is chosen for subsequent burst superimposition testing, conducted as per the testing protocol.

**Figure 2 bioengineering-11-00137-f002:**
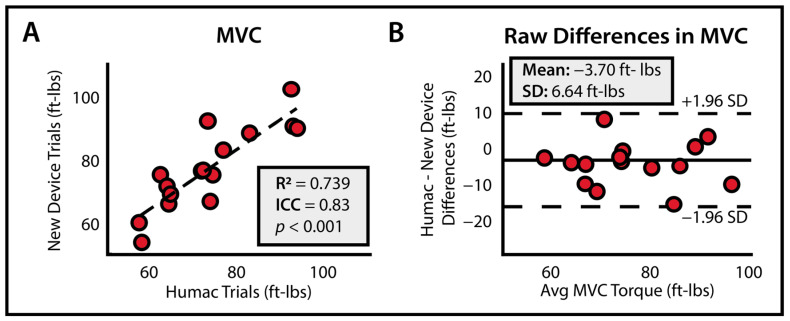
**Accuracy of CEDRS** (**A**) Maximum plantarflexor strength measurements using CEDRS are highly accurate compared to the gold standard with (**B**) small mean differences between devices.

**Figure 3 bioengineering-11-00137-f003:**
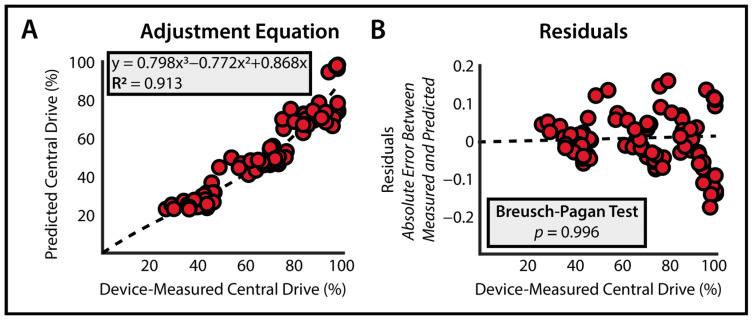
**Validation of Adjustment Equation** (**A**) A third-order polynomial explains most of the variance in device-measured and true central drives. (**B**) There is not a significant difference in residuals across baseline torque values.

**Figure 4 bioengineering-11-00137-f004:**
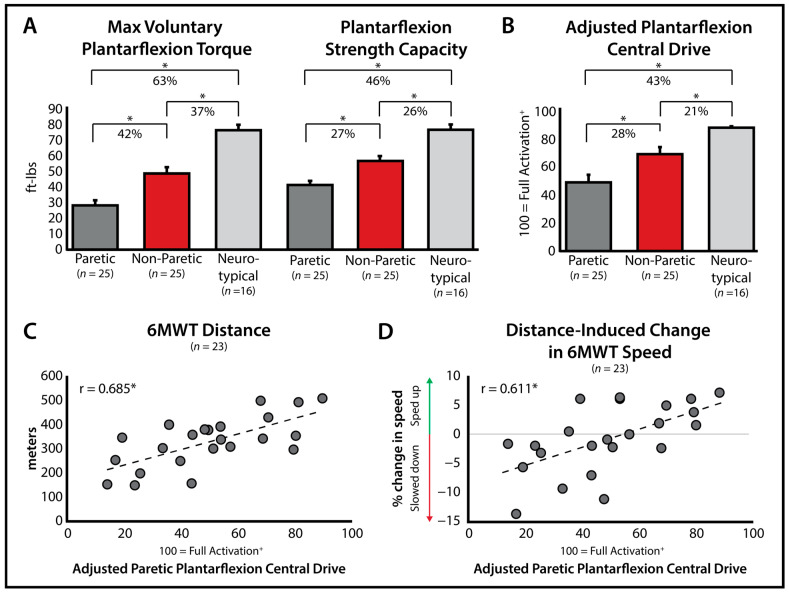
**Post-stroke Validation.** Differences between paretic, non-paretic, and neurotypical limbs for maximum plantarflexor strength and maximum force generating ability (**A**) and central drive (**B**) (*p* < 0.05). Data are presented as the Mean ± SE. Relationships between paretic limb plantarflexion central drive and (**C**) six-minute walk test (6MWT) distance and (**D**) distance-induced %Δ6MWT speed ([min6-min1]/min1; * *p* < 0.05; ^+^ the y-axis in (**B**) and x-axis in (**C**,**D**) presents the range of central drive measurement, from 0 to 100, with 100 indicating full central drive (i.e., full volitional access to the maximum force-generating ability of the plantarflexor muscles)).

**Table 1 bioengineering-11-00137-t001:** Plantarflexion strength variables.

	Neurotypical	Paretic	Non-Paretic	Significance
Fvol (ft-lbs.)	76.21 ± 13.84	27.99 ± 14.15	48.25 ± 18.75	*p* < 0.001 *^,+,#^
MFGA (ft-lbs.)	76.47 ± 13.59	41.07 ± 13.41	56.54 ± 15.71	*p* < 0.001 *^,+,#^
Central Drive	88.73% ± 1.71%	50.39% ± 21.44%	70.12% ± 20.25%	*p* < 0.001 *^,+,#^

* *p* < 0.05 between neurotypical and paretic limbs; ^+^ *p* < 0.05 between neurotypical and non-paretic limbs; ^#^ *p* < 0.05 between paretic and non-paretic limbs

## Data Availability

The data supporting the main conclusions of this manuscript are located within the main text or Appendix A. Additional data are available upon request.

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
