# Peer review of "A Portable, Neurostimulation-Integrated, Force Measurement Platform for the Clinical Assessment of Plantarflexor Central Drive"

_bioengineering, 2024, doi:10.3390/bioengineering11020137_

Round 1
Reviewer 1 Report
Comments and Suggestions for Authors
The study is worthy. The developed device is potentially useful for post-troke functional assessment. At this respect, it would be welcome to discuss what are the potential benefits of using CEDRS with respect conventional dynamometers.
The materials and methods are very difficult to follow as the description of the two experiments are splitted in Procedures, Data Proccessing and Statistical Analysis. It would be more useful to describe all these aspects for experiment #1 and then for experiment # 2.
It would be very welcome to start the description of each of the experiments with the purpose of the experiment.
Lines 184-185, declare that the electrodes are for FES.
Line 191: What does this figure has in common with burst superimposition?
Lines 190-197: Very hard to follow. To this reviewer the procedure is more or less as follows, but not 100% sure (please clarify):
1- Muscle potentiation (how it is performed?)
2 - Selection of optimum pulse width.
2.1 - Twitch stimulations with the muscles at rest
2.2- Measurement of the twitch response (how it is done?).
2.3 - Selection of the pulse width: The minimum pulse width generating the largest twitch response.
3 - Ask the participant to perform maximum plantarflexion torque.
4 - Activate FES while a visual steady state has achieved in the plantarflexion torque with the optimal pulse width.
Has pulse duration been optimized for post-stroke participants?
It is not 100% clear for this reviewer the role of the FES beyond defining the MFGA.
Lines 190-198: In which device is selected the optimal pulse with? (CEDRS or dynamometer?)
Lines 273-274: First comment to central drive tests (prior to description?)
Line 280: 6MWT is in "Data Processing" and should be in "Study Procedures"
Line 283: Not sure what it is the "distance-induced change in xis-minute walk test speed"
Lines 352-353: Confidence intervals for the coefficients of the equation would be welcome and can be obtained from the leave-one-out method.
Lines 392-...: I don't understand what is "Combining the two participants without 6MWT, etc...)
Figure 4C and Figure 4D: Whats the meaning of "100 = Full Activation"?
Author Response
Please see attachement

Reviewer 2 Report
Comments and Suggestions for Authors
Very well written manuscript
Only one comment, L 39 - would use () to go around your i.e.,
You do that in other instances in the paper
Reviewer 3 Report
Comments and Suggestions for Authors
Review of work:
''A portable, neurostimulation-integrated, force measurement platform for the clinical assessment of plantarflexor central drive''
1.The Materials and Methods need considerable improvement. Firstly, please switch the order. Information about patients, recruitment etc. should come before the research tool.
2.Figure 1. - Remove outer black frame.
3.Justify sample size - add sample size calculations.
4.L175 – ‘’ no known neurological impairments’’. How has this been verified? Specify what you mean by '' impairments''. A description of the exact exclusion criteria is missing. The authors are investigating a plantarflexor force and central drive measurement system.
5.The text shows that the group was not heterogeneous. Why did the authors choose to study such a small group not heterogeneously?
6.There is a lack of statistical comparison of numbers, age, BMI and dominant side between groups. Please add.
7.L313 – ‘’ as small (≥0.2), medium (≥0.5), or large (≥0.8) effects’’ - These are the quantities assumed for Cohen's d for eta square are different. This is a factual error.
It should be ‘’ as small (≥0.01), medium (≥0.06), or large (≥0.14) effects’’
8.The conclusions are too much a repetition of the results and discussions. Please rephrase.
Kind Regards
